# MGMT ProFWise: Unlocking a New Application for Combined Feature Selection and the Rank-Based Weighting Method to Link MGMT Methylation Status to Serum Protein Expression in Patients with Glioblastoma

**DOI:** 10.3390/ijms25074082

**Published:** 2024-04-06

**Authors:** Erdal Tasci, Yajas Shah, Sarisha Jagasia, Ying Zhuge, Jason Shephard, Margaret O. Johnson, Olivier Elemento, Thomas Joyce, Shreya Chappidi, Theresa Cooley Zgela, Mary Sproull, Megan Mackey, Kevin Camphausen, Andra Valentina Krauze

**Affiliations:** 1Radiation Oncology Branch, Center for Cancer Research, National Cancer Institute, NIH, 9000 Rockville Pike, Building 10, CRC, Bethesda, MD 20892, USA; 2Caryl and Israel Englander Institute for Precision Medicine, Weill Cornell Medicine, New York, NY 10021, USA; 3Department of Neurosurgery, Duke University, Durham, NC 27710, USA; 4National Tele-Oncology, Veterans Health Administration, Durham, NC 27710, USA

**Keywords:** glioblastoma, MGMT, proteomic, protein, feature selection, machine learning, pattern recognition

## Abstract

Glioblastoma (GBM) is a fatal brain tumor with limited treatment options. O6-methylguanine-DNA-methyltransferase (MGMT) promoter methylation status is the central molecular biomarker linked to both the response to temozolomide, the standard chemotherapy drug employed for GBM, and to patient survival. However, MGMT status is captured on tumor tissue which, given the difficulty in acquisition, limits the use of this molecular feature for treatment monitoring. MGMT protein expression levels may offer additional insights into the mechanistic understanding of MGMT but, currently, they correlate poorly to promoter methylation. The difficulty of acquiring tumor tissue for MGMT testing drives the need for non-invasive methods to predict MGMT status. Feature selection aims to identify the most informative features to build accurate and interpretable prediction models. This study explores the new application of a combined feature selection (i.e., LASSO and mRMR) and the rank-based weighting method (i.e., MGMT ProFWise) to non-invasively link MGMT promoter methylation status and serum protein expression in patients with GBM. Our method provides promising results, reducing dimensionality (by more than 95%) when employed on two large-scale proteomic datasets (7k SomaScan^®^ panel and CPTAC) for all our analyses. The computational results indicate that the proposed approach provides 14 shared serum biomarkers that may be helpful for diagnostic, prognostic, and/or predictive operations for GBM-related processes, given further validation.

## 1. Introduction

Glioblastoma (GBM), a highly aggressive primary brain tumor, presents a critical challenge in clinical management due to its significant heterogeneity, poor prognosis, and limited treatment options [1,2]. Management requires maximal surgical resection followed by concurrent chemoirradiation (CRT) with radiation therapy (RT) and temozolomide (TMZ), followed by the administration of adjuvant TMZ [1,2,3,4]. Identifying reliable prognostic markers, such as MGMT promoter methylation status, and harnessing MGMT protein expression levels are crucial for tailoring therapy and potentially improving patient outcomes [5,6].

MGMT (O6-methylguanine-DNA methyltransferase) [7], is a gene that repairs alkylating agent-induced DNA damage [8]. Patients with methylated MGMT promoters (mMGMT) have been found to exhibit significantly higher treatment sensitivity and improved survival [9,10], highlighting the importance of accurate status prediction for personalized treatment options [7,11]. Additionally, MGMT protein expression reflects the functional activity of the repair pathway, providing complementary information for treatment decisions [3]. The capture of MGMT status has variability in technique [12], but is based on tissue specimens. The connection between MGMT status and MGMT protein expression, which can be captured in serum, can be discordant, with no distinct threshold identified for MGMT protein expression [13]. MGMT status is linked to the response to TMZ, which is the standard of care administered concurrently with RT, and adjuvantly; however, this mainstay approach still results in very poor outcomes, with a median survival of 14 months [9]. The biological mechanisms that underlie treatment resistance to TMZ are multifaceted and poorly understood [14]. The linkage of MGMT methylation status and MGMT protein expression has the potential to offer a more comprehensive picture of tumor biology, potentially leading to improved clinical outcomes and offering independent prognostic value [5,6]. Therefore, the accurate prediction of molecules and pathways that connect both aspects holds immense potential for personalizing treatment strategies. This can be achieved by studying new approaches to systemic management, including agents that act to potentially reverse or modify resistance by targeting GBM stem cells [15,16], or modify response to RT [17] in an effort to link systemic management and RT to both MGMT-dependent and MGMT-independent mechanisms.

Proteomics represents the study of large-scale analyses of protein expression [18] with immense potential for uncovering novel GBM biomarkers with high accuracy and clinical utility by providing complementary information to metabolic and genomic data [18]. Compared to tissue acquisition, analyzing proteins in serum or plasma offers a minimally invasive approach [2,18]. By analyzing complex protein profiles, more profound insights are possible through the identification and monitoring of key players in treatment response and resistance mechanisms identified. Recent advancements in proteomics, whether they employed mass spectrometry and tumor tissue or large-scale panels and serum biospecimens, have uncovered crucial relationships between GBM tumor heterogeneity [19], the proteome and the metabolome [20], the proteome and outcomes [21,22], and the observable footprint of alterations in concurrent management in proteomic data [23]. There is currently no serum biomarker in clinical use for GBM diagnosis or treatment monitoring. However, serum proteomic markers of promising importance are emerging in the literature, and several have applicability to GBM [24].

Utilizing feature selection methods and machine learning algorithms on proteomic data offers additional opportunities to refine biomarker discovery and prediction models. By using diverse datasets and effective dimensionality reduction techniques, hidden patterns can be extracted, and the prediction performance for MGMT status and protein expression can be improved. The feature selection process is a significant data pre-processing step and a dimensionality reduction technique to eliminate redundant and irrelevant predictors in high-dimensional data [2,25,26]. This operation decreases the computational time, reduces complexity, extracts hidden data patterns, makes the related analysis more effective, and makes visualization easier [2,26,27]. Feature selection methods can be categorized into filter (e.g., Minimum Redundancy Maximum Relevance (mRMR)), wrapper, and embedded (e.g., Least Absolute Shrinkage and Selection Operator (LASSO)) methods, according to evaluations of feature subsets [28,29,30]. Apart from the feature selection process, feature weighting is also a crucial step in assigning suitable weights to the features for finding the most effective possible final feature subset, after applying each fold of the cross-validation operation task [26,31].

This study investigates the potential to employ a hybrid filter and embedded feature selection (i.e., mRMR and LASSO) and the rank-based weighting method to predict serum protein biomarkers associated with known molecular markers. To extract the most impactful signals from the high-dimensional serum proteomic data, we leverage a feature selection and weighting technique, ultimately yielding a focused possible minimal number of informative selected features for our large-scale oncologic dataset. By providing more accurate predictions of MGMT status and protein expression, this research can potentially improve clinical outcomes in patients with GBM by offering more tailored treatment approaches.

The primary contributions of this study, categorized as technical and clinical aspects, are summarized below.

### 1.1. Technical Aspects

This study stands out as the first to utilize the innovative and hybrid feature selection and rank-based weighting methodology (i.e., MGMT ProFWise) for both MGMT promoter methylation status classification and MGMT protein expression level regression tasks on proteomic data.

To validate the MGMT ProFWise method and identify shared biomarker features across diverse tasks and datasets, we apply it to two distinct proteomic datasets.

To compensate for the skewed class distribution in our dataset, we employed stratified cross-validation during machine learning training, ensuring each fold accurately reflected the overall class imbalance.

We also handle the identification of the names of the final feature subset after obtaining possible different feature subsets of each cross-validation fold for the feature selection process by employing a rank-based feature weighting procedure.

We explore the impact of the feature selection and rank-based weighting method on the performance of diverse learning models.

We aim to identify the best machine learning model with the minimum number of selected features providing the best performance (i.e., accuracy rate (ACC) or mean squared error (MSE)) for the relevant classification and regression tasks on large-scale proteomic datasets.

We give a general alteration analysis for the selected features of our local proteomic dataset regarding MGMT status.

### 1.2. Clinical Aspects

This study offers encouraging findings to advance GBM serum proteome biomarker research.

We investigate the associations between the identified features, GBM, relevant signaling pathways, and upstream regulatory proteins driving these pathways through STRING applications.

We also analyze shared selected features between MGMT promoter methylation classification and the MGMT protein expression feature selection processes towards ease of transferability for bioinformatics researchers in this domain.

The subsequent sections of this study are organized as follows: Section 2 describes the results, including experiments, performance metrics, and comprehensive computational results. Section 3 presents the discussion of the results. Section 4 is comprised of the materials and methods, including the dataset description and characteristics, and an overview of the utilized feature selection and weighting methodology, and also explains the related feature selection methods and supervised learning models for both classification and regression tasks. Section 5 concludes this study and proposes potential avenues for future research.

## 2. Results

In this section, we describe the experimental processes and evaluation metrics utilized, while also presenting the comprehensive computational results in the following subsections.

### 2.1. Experimental Process

To implement the proposed methods in this study, we employed Python’s scikit-learn [32] library for machine learning tasks and the mRMR [33] package for a filter-based feature selection process. All experiments were conducted on a macOS Ventura MacBook Pro (2.3 GHz 8-core Intel Core i9, 16 GB of 2667 MHz DDR4 RAM), manufactured by Apple, sourced in NIH/NCI/ROB, Bethesda, MD, USA. To achieve optimal results, this study employed five diverse predictive models, including Support Vector Machines (SVM), Logistic Regression (LR), K-Nearest Neighbors (KNN), Random Forests (RF), and AdaBoost, which were used in both feature selection and classification tasks. We adopted the default parameter settings for each classifier involved in the study (i.e., SVM employed C = 1, a radial basis function kernel, and automatic gamma scaling. KNN considered the five nearest neighbors with the Minkowski distance metric and uniform weights. LR utilized an L2 penalty with a C value of 1.0. RF constructed 100 trees using the Gini impurity criterion. AdaBoost generated 50 estimators with the SAMME.R algorithm and a learning rate of 1.0). To control for randomness and to guarantee repeatable results on the proteomic datasets, we set the random state to zero for all five learning models.

To minimize potential bias stemming from the feature selection process due to the different range of each feature value, we employed normalized and logarithmic two-base transformations of feature values on proteomic data obtained from SomaLogic aptamer-based SomaScan^R^ assay technology [34] for our local proteomic dataset. Normalized proteomic data values were also used during validation. Data preprocessing, including feature normalization, can significantly affect the effectiveness of feature selection techniques during the preprocessing stage for the prediction tasks. Features with high variance can be misleading for the feature selection algorithms. The feature normalization process helps mitigate this bias by focusing on the features that contribute meaningfully to the classification/regression task. In this study, Somalogic data arrived normalized from the company by default. However, as the normalized data had a high variance, we applied an additional logarithmic two-base transformation to reduce bias and provide more relevant feature selection. With respect to the CPTAC dataset, we adopted one of the most frequently applied data preprocessing approaches (i.e., z-score normalization). We also applied stratified five-fold cross-validation to guarantee that each fold had approximately the same proportion of samples from each class as the whole dataset, mimicking the real-world distribution and reducing bias towards the majority class for the feature selection and weighting processes. With stratified cross-validation, we achieved a more robust validation of our results with respect to real-world datasets [35]. For the feature selection operation, potential biases and confounding factors may include selection bias (e.g., using the same data for the feature selection and classification), class imbalance, or interactions/correlations between features. To tackle these possible issues, we employed a stratified cross-validation technique to reduce bias by ensuring each fold had a similar ratio of classes as the original data, giving a more accurate view of the model’s generalization ability across all labels. This approach provides a more reliable performance metric by ensuring the balanced spread of classes across folds and reflecting the model’s performance on all classes. By ensuring each fold in cross-validation reflects the true proportion of classes in the data, stratified cross-validation guards against evaluation bias and prevents the model from being trained on an unrealistic distribution of classes. This approach also prevents the skewed training of learning models. Due to unknown MGMT status cases, our stratified five-fold cross-validation technique corresponded to a five-fold cross-validation technique for only the local MGMT dataset for the regression task. We also utilized the minimum redundancy and maximum relevance feature selection method to mitigate feature correlation-related bias factors. We evaluated all selected features according to the average performance values of the supervised learning models after using the cross-validation technique. In the feature weighting stage, we also tried all possible rank-based values for two feature selection methods applied, to obtain the best outcome among relevant feature subsets. To determine the final selected feature subset, all possible minimum weight values were tested to obtain the best feature subset with the best prediction performance and the minimum number of selected features. In this study, we adopted the same parameter settings implemented by Tasci et al. [2] for MGMT-based feature selection. For the classification task, MGMT methylated and unmethylated statuses were identified as positive and negative classes, respectively. On the other hand, MGMT protein expression level was predicted from the selected feature subsets for the regression task.

### 2.2. Performance Metrics

To assess the effectiveness of the utilized hybrid filter and embedded feature selection method for MGMT processes, we utilized two task-specific metrics: classification accuracy for classification tasks and mean squared error for regression tasks.

For classification, ACC was calculated by dividing the sum of true positives and true negatives by the total number of samples (including false positives and false negatives) [36], as detailed in Equation (1),(1)$$\mathrm{ACC}=\frac{\mathrm{TP}+\mathrm{TN}}{\mathrm{TP}+\mathrm{TN}+\mathrm{FP}+\mathrm{FN}}$$where TP, FP, TN, and FN denote the number of true positives, false positives, true negatives, and false negatives, respectively.

For regression, MSE is a common performance metric used to assess the performance of a predictor. It measures the average squared difference between the predicted values and the actual values. In simpler terms, it gives information about how far off predicted values are, on average, from the actual values. The related equation for MSE [37] is defined in Equation (2),
(2)$$\mathrm{MSE}\;=\frac{\sum_{i=1}^{n}{\left(y_{i}-\lambda \left(x_{i}\right)\right)}^{2}}{n}$$
where y_i_ is the actual target value for test instance x_i_, λ(x_i_) is the predicted target value for test instance x_i_, and n is the number of test instances.

### 2.3. Computational Results

This subsection explores the effect of feature selection and weighting approaches on the learning models’ performance. We organized this subsection to present MGMT status and MGMT protein expression level-based computational results in detail for our local proteomic and CPTAC [20] datasets separately.

#### 2.3.1. Local Proteomic Dataset-Based Results

##### Impact of Feature Selection Methods on Classification Model Performance for MGMT Status

The mean performance results of five different machine learning models with or without the effects of applying feature selection methods using stratified cross-validation for MGMT known status classification on the local proteomic dataset are illustrated in Table 1. Table 1 indicates that the mRMR method yields better accuracy values for four machine learning models, except for the random forest model, after applying feature selection. # represents number. The mRMR method also produces higher performance results than the LASSO FS method for all models. Before the FS process, generally, the performance values are lower than after the FS operation, except for the random forest model.

##### Impact of Feature Weighting and Selection Methods on Classification Model Performance for MGMT Status

When applying rank-based feature weighting and selection methods, we evaluated all different rank-based weights (i.e., 1 and 2) for the LASSO and mRMR feature selection methods, respectively. The computational results are given in Table 2 in detail, and k represents the minimum number of weights, # indicates number. The best result in Table 2 was obtained with an accuracy rate value of 87.692%, by utilizing the logistic regression model, 60 selected features, and the minimum number weight of two. Thanks to this methodology, sixty features were selected from 7289 human proteomic markers, with respect to the minimum number of features and the highest accuracy rate. Two proteomic markers between 60 features (i.e., sequence ids) have the same ENTREZ gene symbol abbreviations. The logistic regression model provided higher accuracy values than the support vector machine, k-nearest neighbor, random forest, or AdaBoost classifiers for all rank-based weighting values. Considering the results of Table 1 during the cross-validation process, we obtained higher performance values when the mRMR FS method was assigned more weight (i.e., significance level) than the LASSO FS method.

The related quiver plot for each mean feature value is shown in Figure 1, illustrating significant alterations that define the MGMT methylated and unmethylated statuses. Features with higher alterations, including positive and negative values for MGMT known statuses, are marked with blue circles, with the number adjacent to the molecule indicating its rank in the results. The X-axis represents mean feature values for all patients with MGMT unmethylated status. The Y-axis represents mean feature values for all patients with MGMT methylated status. Numbers indicate rank in the analysis. Blue boxes and blue circles indicate the proteins that are most different between one state and another. The grey bar indicates the proteins with a difference between −0.3 and 0.3, between MGMT methylated and unmethylated patients employing pre-CRT serum data transformed for analysis. Heatmap and correlation matrices for the mean values of the selected features, with respect to the MGMT status of our local proteomic dataset, are also provided in Appendix A.

##### Impact of Feature Selection Methods on Regression Model Performance for MGMT Protein Expression Level

The mean performance results for the two different commonly used regression models are presented in Table 3. These are shown with or without the effects of applying feature selection methods using stratified cross-validation for MGMT, all (i.e., known and unknown methylation) protein expression level regression tasks for 109 patients in terms of mean squared error. The mRMR method yields better (i.e., lower) MSE results for support vector regression (SVR) and random forest (RF) regressors than the LASSO FS method, after applying feature selection. The mRMR method produces lower error values than the results without feature selection. The best result was obtained with the MSE value of 0.274, by employing the mRMR FS method and the RF regressor model.

##### Impact of Feature Weighting and Selection Methods on Regression Model Performance for MGMT Status

We computed the effects of rank-based feature weighting and selection method for 109 patients, according to the MGMT all protein expression level-based local proteomic dataset (see Table 4). For all rank-based weights, the related comprehensive computational results are given in Table 4, and k represents the minimum number of weights, # indicates number. The best result was provided with the MSE value of 0.173 and the selected minimum number of features as 19. The regression model was employed as SVR, by assigning weights to the LASSO and mRMR FS methods of 2 and 1, respectively. We also obtained the same and the best MSE value with the SVR model and the selected number of features as 30, by assigning weights to the LASSO and mRMR FS methods of 1 and 2, respectively. We set the number of selected features for optimization (i.e., dimensionality reduction) to 19.

#### 2.3.2. CPTAC Proteomic Dataset-Based Results

##### Impact of Feature Selection Methods on Classification Model Performance for MGMT Status

The mean performance results of five different prediction models with or without the effects of applying feature selection methods using stratified cross-validation for MGMT known status classification on the CPTAC proteomic dataset are shown in Table 5. The mRMR and LASSO FS methods provide generally similar results for five different classification models, and the results before and after applying feature selection vary depending on the FS methods and learning models used.

##### Impact of Feature Weighting and Selection Methods on Classification Model Performance for MGMT Status

After applying rank-based feature weighting and selection methodology for the validation dataset, we assessed all possible different rank-based weights (i.e., 1 and 2) for the LASSO and mRMR feature selection methods, respectively, (see Table 6), with k denoting the minimum number of weights, and # indicating number. The best result in Table 6 was obtained with an accuracy rate value of 90%, by utilizing the support vector machine model, 114 selected features, and the minimum number weight of 2, by assigning weights to the LASSO and mRMR FS methods of 2 and 1, respectively.

##### Impact of Feature Selection Methods on Regression Model Performance for MGMT Protein Expression Level

The mean performance results of two different regression models with or without the effects of applying feature selection methods using stratified cross-validation for the MGMT protein expression level regression tasks of 90 patients in the CPTAC proteomic dataset, in terms of mean squared error, are presented in Table 7. The lowest MSE had a value of 0.772, which was obtained using the LASSO method and support vector regression (SVR) model, performing better than the mRMR FS method after applying feature selection. SVR also produced better performance values than not applying FS methods.

##### Impact of Feature Weighting and Selection Methods on Regression Model Performance for MGMT Protein Expression Level

After using our feature weighting and selection scheme, we also observed the comprehensive effects of the rank-based weighting policy for 90 patients according to the MGMT protein expression level-based CPTAC proteomic dataset in Table 8. For all rank-based weights, k shows the minimum number of weights, # indicates number. The best result was provided with the MSE value of 0.406 and the selected minimum number of features as 56. The employed regression model was SVR, and the assigned weights to the LASSO and mRMR FS methods were 2 and 1, respectively. This rank-based weighting scheme also supported the significance level effect of the LASSO FS method, seen in Table 7. The SVR model yielded better MSE values for all rank-based weights than the RF regression model.

We also obtained experimental results in detail, according to the MGMT known protein expression level-based local proteomic dataset (Appendix A) and present all selected features with respect to Entrez Gene Symbols in Appendix A.

## 3. Discussion

MGMT promoter methylation remains the cardinal molecular feature associated with survival in GBM, rendering it both a prognostic and predictive biomarker [9,10]. The improvement in outcomes observed in MGMT promoter methylated patients has been attributed to its role as a repair enzyme and its activity in relationship to the administration of TMZ, specifically for GBM [13]. However, its action in direct DNA repair, which constitutes less than 10% of the damage inflicted by TMZ [14,38], as well as its relevance to the presence of RT alone [39] and to other malignancies, and in the prevention of cancer development [40] suggest a broader role of the MGMT promoter. Further, the outcome is linked to several facets, including methylation patterns of CpGs associated with prognosis [41].

In this study, we employed MGMT promoter status, serum MGMT expression measurements from a large-scale proteomic panel, and ML feature engineering to identify serum signals that capture the relationship between MGMT promoter status and MGMT protein expression. To our knowledge, this is the only study that has attempted this linkage through employing serum MGMT protein expression. Our hypothesis is that, with growing serum data availability and the interpretable use of ML and feature engineering, serum signals can be effectively harnessed in specific disease entities such as glioma.

We showcase several technical innovations employed for the purpose of linking MGMT status and MGMT protein expression to molecules measured in the serum of patients, post-surgical resection but prior to upfront management with CRT. The technical approaches presented here carry broad transferability to other molecular domains and have wide dimensionality for large-scale data. In our previous work, we identified features associated with the administration of CRT [2]. More features were shared between MGMT status and CRT administration than by MGMT protein expression and CRT administration in this study (Figure 2, Appendix A).

The proteins identified have relevance to GBM. PLA2G12B, a phospholipase A2 variant, is of interest, given recent evidence for the role of phospholipases in cancer and documented high expression levels in glioma, as well as its emergence as an oncogene associated with glioma progression [20,42]. ABO, histo-blood group ABO system transferase, poses an interesting link between MGMT status and blood group. The blood group has been examined in connection with several conditions, including GBM, but there is currently no conclusive evidence [43]. The most shared features (10) (Figure 2) between the MGMT analysis and the CRT analysis in the local dataset have reported relevance to GBM: ADGRF2 (overexpressed in glioma, regulating proliferation and migration) [44], AHSG (serum level predicts survival in GBM) [45], CRP [46], ME2 (promotes proneural mesenchymal transition) [47], MMP1 (expression upregulated in GBM and associated with decreased survival) [48], Sigle9 (immune evasion, reduced survival) [49], and others which are not yet fully defined: FCGR3A, LAT, and PRTN3 [20]. We have previously identified these signals as associated with CRT [2] and MGMT expression, and associated with lower survival [21].

Serum MGMT expression and CPTAC notably overlapped with respect to CTSA (upregulated in glioma and associated with immune infiltration) [50], MTFHD1 (one-carbon metabolism association in GBM) [51], and CD320 (implicated in a cobalmin-mediated metabolism) [52]. The shared features are linked in STRING (Figure 3) with MGMT protein expression, in nodes that are connected to glucose, folate, and a one-carbon metabolism, as well as stress response, radio, and chemoresistance in GBM [53]. MGMT protein expression, however, only emerged as a shared feature between CPTAC and our previous study through examining features associated with CRT in the local dataset [2]. This may reflect the inconsistent or low capture of MGMT protein expression levels in serum. While mean MGMT protein expression was higher in unmethylated patients vs. methylated patients, overall MGMT expression levels carried significant overlap between MGMT methylated and unmethylated patients in both the local dataset and in CPTAC (Figure 4), which was similar to previously reported trends in the literature [20]. This study, however, is encouraging, since MGMT protein expression in serum can be linked to MGMT protein expression in tissue via mechanistic connections wherein each method of measurement (MGMT promoter status, MGMT protein expression) more optimally captures a specific mechanistic aspect vs. another. This study also importantly illustrated that features associated with the administration of CRT for GBM are also associated with MGMT status. This can potentially extend our understanding of how MGMT promoter methylation may be exerting wide-ranging effects to result in improved outcomes in patients with MGMT methylated disease. Large-scale proteomic panels are actively evolving to include larger repertoires of proteins; thus, as datasets grow and validation of findings improves, ranges can be generated for various GBM clinical feature sets, and markers displayed in Figure 2B can be potentially employed to predict MGMT status. Currently, MGMT status is determined based on tissue sample obtained at the time of surgical resection, the analysis of which can be cost prohibitive and delay results which are often not available at the time of initial consult (2–3 weeks post resection). Serum markers can be measured with results returned within hours, as is currently the case with markers such as CRP, which is one of the markers identified in the current analysis. For treatment, monitoring panels can be finetuned further to home in on patterns that reflect the likelihood of response or treatment failure, rendering the patient a candidate for clinical trial management. For example, serum markers that phenotypically reflect tumor behavior that mirrors the unmethylated patients in Figure 1 could be considered more optimal candidates for treatment with agents other than TMZ [15,54]. Limitations of this study include the large period over which patients were diagnosed and treated, and the difficulty in comparing protein signal originating in serum to protein signal measured in tissue. MGMT status was established through different methods in this study vs. in CPTAC data [20]. Future directions of our research include the validation and clinical translation of serum biomarkers into clinical trials. Validation is subject to comparative analyses with serum and plasma proteomic datasets, several of which are currently evolving [34]. Clinical translations are contingent on the implementation of serum markers in GBM trials. While over 30 trials are ongoing that aim to leverage biomarkers for the diagnosis or management of GBM [55], most involve tissue as the biospecimen of origin. A shift to liquid biopsies is needed; however, a template for the real-time measurement of novel serum markers needs to be established for analysis to occur in the clinic to direct management in real-time, and this is the subject of future directions.

## 4. Material and Methods

In this section, we explain the patient characteristics and give the main information in our local dataset and validation dataset. Then, in the following subsections, we describe our methodology and present a general overview of employed terms, methods, and prediction models, accordingly.

### 4.1. Datasets

We employed two proteomic datasets to select features with respect to classification and regression tasks for this study. The first proteomic dataset is our local dataset comprised of 109 patients diagnosed with pathology-proven glioblastoma (GBM) between 2005 and 2023. All patients underwent upfront CRT. Serum samples were collected on the study before initiation of CRT (average 6.7 days, range (0 to 24)) and after completion of CRT (average 0.33 days, range (−1 to 31)) [21,56]. The average time between pre- and post-sample acquisition averaged 48 days (ranging from 22 to 83 days) [56]. After collection, the serum samples were frozen at −80 °C for an average duration of 3951 days (with a range of 239 to 7072 days) [56]. Subsequently, the samples were thawed and screened using the aptamer-based SomaScan^®^ [34] proteomic assay. This assay employs a multiplexed, aptamer-based approach to measure the relative concentrations of 7596 protein targets (including 7289 human proteins) for changes in expression [2], using approximately 150 μL of serum. The patient characteristics for our local proteomic dataset are illustrated in Table 9. The dataset-storing operations were provided by the NIDAP environment [57].

For the validation task, we utilized the Clinical Proteomic Tumor Analysis Consortium (CPTAC) [20] proteomic dataset to select proteomic markers as well. Our first aim was to identify the relevant and significant biomarker sets regarding MGMT status or expression level from the large-scale proteomic dataset. To carry this out, we employed our local MGMT-based proteomic dataset. Given the paucity of MGMT-based proteomic datasets, we also utilized CPTAC as a parallel proteomic dataset to validate our results and identify the shared biomarkers while methodologically approaching the similar tasks. The local proteomic dataset was the result of biospecimens collected on trial at our institution. The CPTAC proteomic dataset was selected for parallel analysis for the following reasons: (1) it is currently the single most accessible proteomic dataset that is also linked to The Cancer Imaging archive; (2) it has MGMT status as an available feature; (3) it is the proteomic dataset most popular in publications in this space. These facets were central to its selection, as future directions include the linkage to imaging features for our group. Thus, the selection of a set that was already linked to imaging data was central. We were equally keen to select a comparison set that could, through linkage to serum data, most expeditiously advance, evolving research given ease of accessibility and widespread use in the literature to eventually transfer findings. MGMT protein expression corresponding to MGMT status for our local dataset and the CPTAC dataset based on RFU and normalized data is shown in Figure 4. While our local dataset contains protein measurement values in serum with RFU data, the CPTAC dataset indicates the protein measurement in tissue with normalized data values for MGMT known statuses (i.e., methylated and unmethylated). The main characteristics of the feature selection process on both these two datasets are also illustrated in Table 10 and Table 11.

### 4.2. Methodology

In this section, we provide a general overview of the proposed scheme, describe the utilized feature selection and weighting methodology, and briefly explain the related classification and regression models.

#### 4.2.1. Proposed Scheme

We adopted a hybrid method for the MGMT-based feature selection process, combining ranking-based feature weighting and filter and embedding-based feature selection methods [2,26] in this study. The mRMR and LASSO methods are among the most commonly used and important algorithms for the feature selection process in the most highly cited papers in the literature [58,59]. In this study, we utilized the advantages (i.e., power) of both different types of feature selection methods (i.e., filter and embedded) by combining two different algorithms via a rank-based weighting methodology. We also adopted five effective and popular machine learning classification models [60], including SVM, LR, KNN, RF, and AdaBoost, to assess the performance of our feature selection methodology. This two-phase feature selection approach involves:

**Feature selection:** We employed hybrid LASSO and mRMR techniques to identify relevant features from the datasets based on their importance level and redundancy.

**Feature weighting:** We tried different rank-based weights to assign values to each selected feature set for both feature selection methods, reflecting their significance for classification or regression tasks [2,26].

The details of this hybrid approach, including the algorithmic diagram and specific processes, are presented in Figure 5.

The feature selection starts with feeding all proteomic markers into a feature selection process using stratified cross-validation to reflect the class imbalance of the dataset. For each data fold, the features selected by both feature selection methods are recorded, and their counts are increased based on their assigned weights from the ranking procedure [2,12]. Next, the feature list with the lowest overall weight is evaluated for all possible weight values. Finally, the final set of features is selected by evaluating all weight combinations and identifying those that achieve the best performance value. In this research, we also tried different rank-based weights to achieve the best performance score for the related dataset and operation. A detailed explanation of this approach can be obtained from Tasci et al. [2,26].

#### 4.2.2. Feature Selection Methods

Feature selection, also known as variable or attribute selection, automatically identifies and selects the most relevant attributes from the dataset by reducing dimensionality, providing faster and more cost-effective operations, better model interpretation, and/or improving prediction performance [2,25,26,30]. The selected features can contribute significantly to the performance of the predictive model used. Feature selection methods are generally grouped into three categories: filter, wrapper, and embedded methods, based on the evaluation criteria of the feature subset. While filter feature selection methods evaluate features independently of the model (e.g., mRMR), wrapper methods assess subsets of features by training and evaluating models iteratively. Embedded methods combine the feature selection process within the model training process (e.g., LASSO) [2,26]. In this study, we adopted both hybrid filter and embedded feature selection methods, as used by Tasci et al. [2,26].

##### mRMR

The mRMR method, known as Minimum Redundancy Maximum Relevance (mRMR), is a filter-type feature selection algorithm used in the context of pattern recognition and machine learning. This method aims to provide a balance between selecting features that are highly correlated with the target or class label (i.e., maximum relevance) and avoiding redundancy (i.e., low correlation) between selected features [61].

The mRMR FS method assumes that we have a set of m features in total. Each feature, called Xi (i ∈ {1,2,3, …, m}), has an importance score based on the mRMR method. This score is calculated using Equation (3) [2,62,63],
(3)$$f^{\mathrm{mRMR}}\left(X_{i}\right)=I\left(Y,{\;X}_{i}\right)-\frac{1}{\left|S\right|}\sum_{X_{s}\in S}I\left(\mathrm{Xs},\mathrm{Xi}\right)$$
where Y is the target class variable, S is the selected features set, |S| is the number of features, Xs ∈ S is one feature out of feature set S, and Xi shows a feature currently not chosen: Xi ∈ S.

At each step, the mRMR FS method picks the most informative feature and adds it to the chosen set. This “informative” selection means the feature is both relevant to the target variable and different from the ones already chosen.

##### LASSO

LASSO, an abbreviation for Least Absolute Shrinkage and Selection Operator, is an embedded feature selection method used for machine learning and regression analysis. LASSO introduces a penalty term in the model training process to encourage sparse solutions, effectively shrinking the coefficients of irrelevant or less important features towards zero. By controlling the strength of the penalty term, LASSO performs both feature selection and regularization, promoting automatic feature subset selection. LASSO’s ability to handle high-dimensional datasets makes it particularly useful in scenarios where the number of features exceeds the number of samples. With LASSO, the selected features tend to exhibit stronger predictive power and enhanced interpretability, allowing for more efficient and accurate modeling. The details of the LASSO feature selection can be found in [64]. Assume X = [x_1_, …, x_p_] to be the feature matrix, and that the data are standardized [64]. The coefficients of a linear model estimated by LASSO are provided by Equation (4) [64]:(4)$$\hat{\beta }=\underset{\beta }{\arg\min}{\left\|y-\sum_{j=1}^{p}{x_{j}\beta }_{j}\right\|}^{2}+\lambda \sum_{j=1}^{p}\left|\beta_{j}\right|\;$$

The LASSO regularization parameter, denoted by λ, plays a crucial role in the LASSO method. Meanwhile, $\hat{\beta }$ represents an unbiased prediction of the degrees of freedom associated with LASSO. Leveraging this insight, we can create an adaptive model selection criterion to efficiently choose the optimal LASSO features [2,25,64].

#### 4.2.3. Feature Weighting Process

In the feature weighting stage, the significance of each selected feature in distinguishing pattern classes is typically denoted by a weight value. This weight can be either added to or multiplied by the feature values as 1 or 2, depending on the performance level of the feature selection methods (i.e., LASSO and mRMR) employed. In this study, we adopted a rank-based feature weighting approach. Specifically, we ranked two feature selection methods—LASSO and mRMR—based on their performance in terms of accuracy rate or mean squared error for the classification and regression tasks, respectively. The more effective FS method received a higher weight compared to no feature selection results. For each fold, we assigned a weight value of 2 or 1 to the selected features, depending on the relevant feature selection method, contributing to the total weight list for LASSO and mRMR. We also tried all different rank combinations for each feature selection method to achieve the best possible outcome and provide a contribution for the prediction tasks on the proteomic datasets for this study.

### 4.3. Classification and Regression

The classification process is a fundamental task in machine learning that involves assigning predefined labels or categories to input data points based on their features. The goal of classification is to build a predictive model that can accurately classify new instances into their appropriate classes. Classification algorithms learn patterns and relationships from labeled training data, enabling them to make predictions on unlabeled data. Commonly used classification algorithms include k-nearest neighbors, logistic regression, support vector machine, random forests, and AdaBoost. We briefly describe these learning models in Appendix A.

The regression process is a type of prediction for continuous values (i.e., to estimate a dependent variable) based on the values of independent variables. There are some common regression algorithms in the literature, such as support vector machines and random forest regression models. The main difference between these processes is to predict discrete (i.e., categorization or classification) or continuous values (i.e., regression) based on the predictors.

#### 4.3.1. Support Vector Machine

Support vector machines (SVMs) are popular supervised machine learning algorithms for classification and regression tasks. SVMs aim to find an optimal hyperplane that separates different classes in the input feature space, maximizing the margin between the classes [65,66]. SVMs can handle both linearly separable and non-linearly separable data by using various kernel functions, such as linear, polynomial, radial basis function (RBF), or sigmoid functions [66,67,68]. SVMs are effective in handling high-dimensional data and are known for their ability to generalize well to unseen data, reducing the risk of overfitting.

#### 4.3.2. Logistic Regression

Logistic regression is a popular classification algorithm used to model the relationship between input features and a binary or categorical target variable, providing probabilistic predictions for each class [69]. The logistic regression model applies the logistic function (also known as the sigmoid function) to map the linear combination of input features to a probability value, which is then used to make class predictions [70]. Logistic regression is a linear model that estimates the coefficients of the input features through maximum likelihood estimation, optimizing the log-likelihood function [71]. It is a well-established and interpretable algorithm that can handle both binary and multi-class classification problems by using techniques like one-vs-rest or softmax regression [72].

#### 4.3.3. K-Nearest Neighbors

The k-nearest neighbors (KNN) classifier is a popular instance-based machine learning algorithm that classifies new data points based on their similarity to the k-nearest neighbors in the training dataset [73]. KNN has been widely used in various domains, such as pattern and image recognition, text classification, and recommendation systems, due to its simplicity and effectiveness [74].

#### 4.3.4. Random Forest

The random forest model is an ensemble learning method that combines multiple decision trees to make predictions. Each tree in the random forest is trained on a random subset of the training data and features, resulting in a diverse set of models that work together to improve prediction performance and handle complex datasets [75]. The computational efficiency and ease of parallelization make it a highly efficient and versatile classification or regression model, which is also particularly resilient to outliers and overfitting [2,76].

#### 4.3.5. AdaBoost

The AdaBoost (Adaptive Boosting) classifier is a powerful ensemble learning algorithm that combines weak classifiers to create a strong classifier. It iteratively adjusts the weights of training samples based on their classification performance, allowing subsequent weak classifiers to focus on challenging instances and improve overall prediction accuracy [77]. Its ability to handle complex datasets and improve weak learners’ performance has made it a popular choice in practical applications [78].

## 5. Conclusions and Future Work

This research presents MGMT ProFWise, a novel method to select and weigh features associated with MGMT promoter methylation status and MGMT protein expression, employing both a local serum-based and a public tissue-based proteomic dataset. MGMT ProFWise combines two established methods (LASSO and mRMR) to identify the most informative features. In other words, several technical innovative approaches are combined in this study to connect promoter status to protein expression and validate findings with publicly available data (CPTAC). From a clinical standpoint, several molecules were identified that are relevant to GBM biology and reveal mechanistic connections that connect MGMT status and expression to metabolic pathways, including glucose and folate metabolism. These findings merit further exploration with metabolomic analysis and validation in larger datasets. The serum biomarkers emerging in this analysis may be helpful for diagnostic, prognostic, or predictive operations for GBM-related processes. Future directions of this study include the addition of metabolomic and imaging features to enhance the identification of non-invasive biomarkers for GBM.

## Figures and Tables

**Figure 1 ijms-25-04082-f001:**
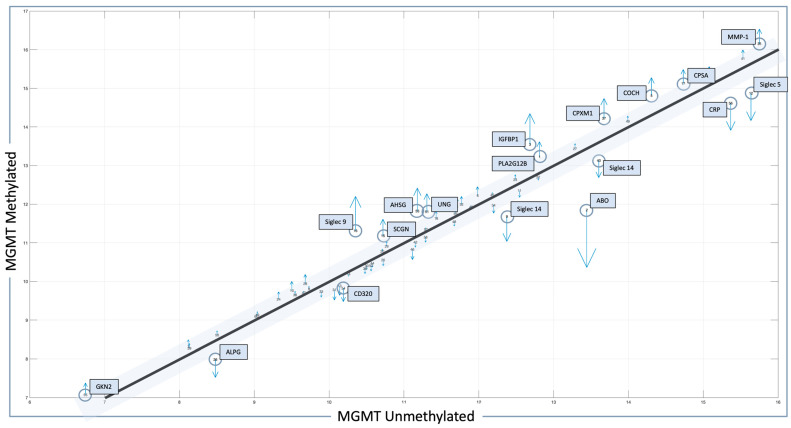
Features identified in the local dataset in association with MGMT status. Protein signals whose level is statistically significant and different between methylated- and unmethylated-status patients are outlined in blue boxes, with the most altered molecules marked with blue circles. Blue arrows indicate the difference in protein levels between MGMT methylated and MGMT unmethylated cases (up arrow = higher, down arrow = lower).

**Figure 2 ijms-25-04082-f002:**
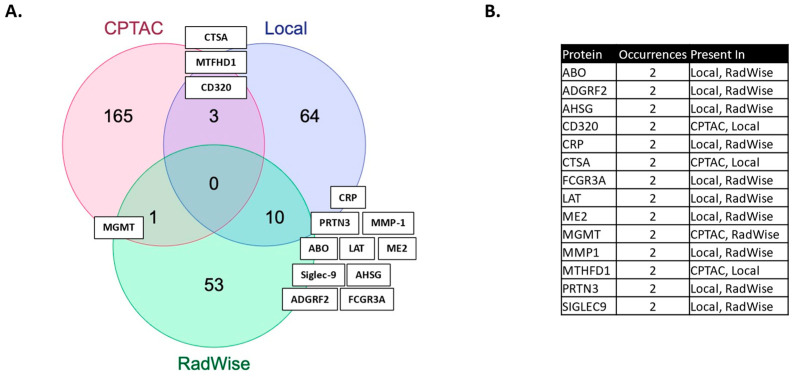
(**A**) Features shared between MGMT status and MGMT protein expression in the local dataset (MGMT local), CPTAC and features shared with the administration of chemoirradiation in the local dataset (RadWise 109 Pts FS). (**B**) Shared protein features between local dataset (CPTAC and CRT) administration.

**Figure 3 ijms-25-04082-f003:**
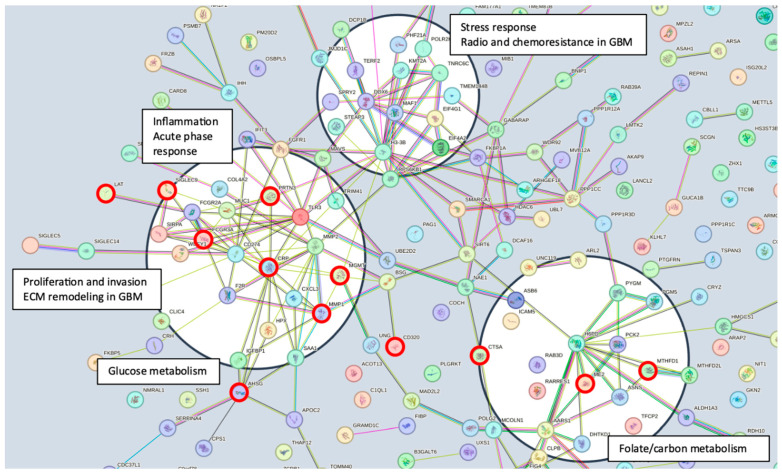
STRING pathway illustrating the mechanistic framework emerging from the inclusion of all identified proteins (https://string-db.org/cgi/network?taskId=buN2PVzYtswE&sessionId=bWFEyxD8L7d7, accessed on 18 January 2024). Several nodes are identified (black circles), linking identified proteins and the MGMT protein to potential mechanisms of biological activity. Connecting lines represent data evidence across the full STRING network. Proteins identified that are shared between analyses (Appendix A, Figure 2) are marked with red circles.

**Figure 4 ijms-25-04082-f004:**
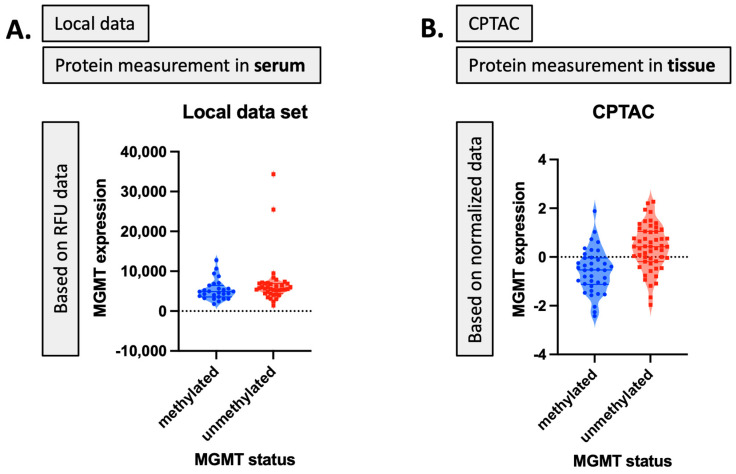
MGMT protein expression corresponding to MGMT status: (**A**) in the local dataset, (**B**) in the CPTAC dataset.

**Figure 5 ijms-25-04082-f005:**
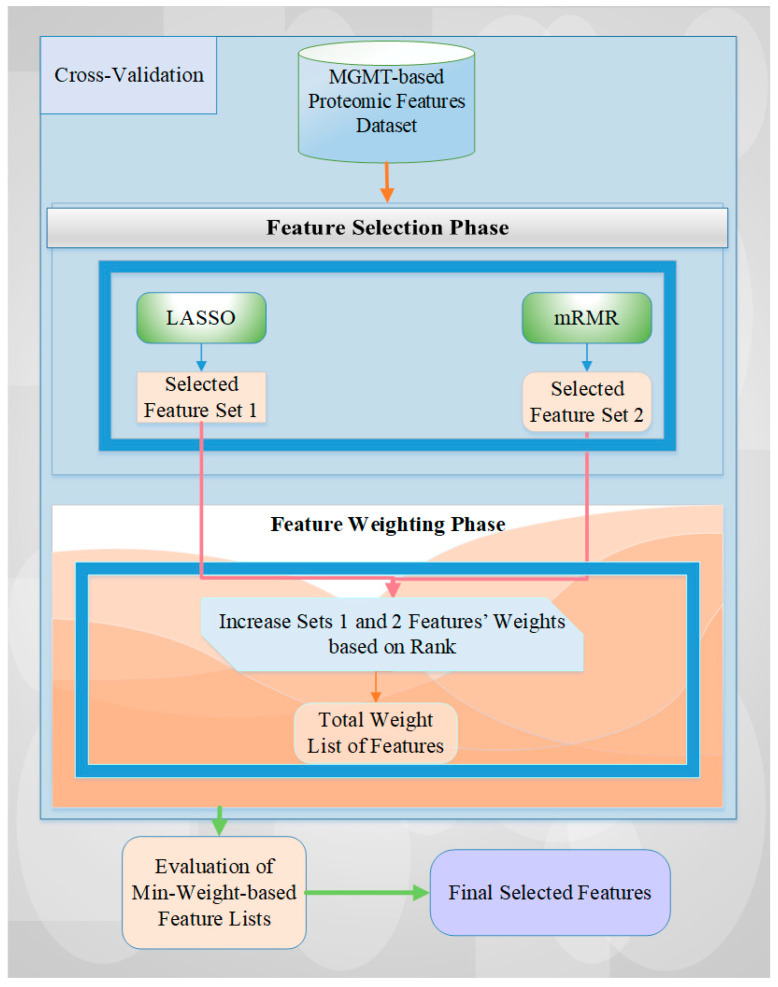
A general overview of the utilized architecture for MGMT-based feature selection.

**Table 1 ijms-25-04082-t001:** The effects of feature selection methods in terms of accuracy rate (%) for 65 patients according to the MGMT-based local proteomic dataset.

ML-ACC	Before FS	LASSO	mRMR
**SVM**	58.461	55.384	64.615
**LR**	61.538	60.000	66.154
**KNN**	56.923	56.923	67.692
**RF**	69.231	52.308	60.000
**AdaBoost**	56.923	50.769	66.154

**Table 2 ijms-25-04082-t002:** The performance results of the rank-based feature weighting and selection method in terms of accuracy rate (%) for 65 patients according to the MGMT known status-based local proteomic dataset. Bold values highlight the best outcome for each method.

**Local Proteomic Dataset**						
**MGMT Known Status-Based FS**						
**LASSO = 1 and mRMR = 2**						
**k**	**# of Features**	**SVM**	**LR**	**KNN**	**RF**	**AdaBoost**
**1**	66	80.000	84.615	73.846	75.385	80.000
**2**	**60**	78.462	**87.692**	70.769	76.923	76.923
**3**	9	76.923	84.615	81.538	67.692	75.385
**4**	9	76.923	84.615	81.538	67.692	75.385
**5**	3	66.154	75.384	70.769	67.692	56.923
**6**	2	67.692	70.769	64.615	69.231	55.385
**7**	2	67.692	70.769	64.615	69.231	55.385
**8**	1	70.769	73.846	73.846	58.461	60.000
**Local Proteomic Dataset**						
**MGMT Known Status-Based FS**						
**LASSO = 2 and mRMR = 1**						
**k**	**# of Features**	**SVM**	**LR**	**KNN**	**RF**	**AdaBoost**
**1**	66	80.000	84.615	73.846	81.539	80.000
**2**	19	75.385	76.923	75.385	70.769	69.231
**3**	7	69.231	78.461	64.615	70.769	72.308
**4**	7	69.231	78.461	64.615	70.769	72.308
**5**	2	66.154	72.308	66.154	56.923	50.769
**6**	2	66.154	72.308	66.154	56.923	50.769
**7**	2	66.154	72.308	66.154	56.923	50.769
**8**	1	69.231	69.231	60.000	41.538	41.538
**9**	1	69.231	69.231	60.000	41.538	41.538
**10**	1	69.231	69.231	60.000	41.538	41.538
**11**	1	69.231	69.231	60.000	41.538	41.538

# represents number.

**Table 3 ijms-25-04082-t003:** The effects of feature selection methods in terms of mean squared error (MSE) for 109 patients according to MGMT all expression level-based proteomic dataset.

ML-MSE	Before	LASSO	mRMR
**SVR**	0.301	0.306	0.281
**RF**	0.292	0.286	**0.274**

**Table 4 ijms-25-04082-t004:** The performance results of the rank-based feature weighting and selection method in terms of mean squared error for 109 patients according to the MGMT all protein expression level-based local proteomic dataset. Bold values highlight the best outcome.

Local Proteomic Dataset							
MGMT All Status Protein Expression Level-Based FS							
LASSO = 1 and mRMR = 2				LASSO = 2 and mRMR = 1			
k	# of Features	SVR	RF	k	# of Features	SVR	RF
**1**	170	0.201	0.224	**1**	170	0.201	0.217
**2**	99	0.174	0.222	**2**	144	0.209	0.230
**3**	45	0.177	0.211	**3**	70	0.181	0.210
**4**	30	**0.173**	0.212	**4**	66	0.179	0.209
**5**	16	0.204	0.202	**5**	37	0.175	0.204
**6**	9	0.228	0.218	**6**	34	0.179	0.217
**7**	6	0.228	0.208	**7**	20	0.177	0.210
**8**	6	0.228	0.208	**8**	**19**	**0.173**	0.222
**9**	3	0.281	0.204	**9**	9	0.234	0.244
**10**	3	0.281	0.204	**10**	9	0.234	0.244
**11**	1	0.282	0.374	**11**	1	0.282	0.374
				**12**	1	0.282	0.374
				**13**	1	0.282	0.374

**Table 5 ijms-25-04082-t005:** The effects of feature selection methods in terms of accuracy rate (%) for 90 patients according to the MGMT-based proteomic CPTAC dataset.

ML-ACC	Before FS	LASSO	mRMR
**SVM**	62.223	57.778	56.667
**LR**	51.111	57.778	53.333
**KNN**	56.667	58.889	60.000
**RF**	65.556	60.000	56.667
**AdaBoost**	53.333	60.000	57.778

**Table 6 ijms-25-04082-t006:** The performance results of the rank-based feature weighting and selection method in terms of accuracy rate (%) for 90 patients according to the MGMT known status-based proteomic CPTAC dataset. Bold values highlight the best outcome.

**CPTAC Proteomic Dataset**						
**MGMT Known Status-Based FS**						
**LASSO = 1 and mRMR = 2**						
**k**	**# of Features**	**SVM**	**LR**	**KNN**	**RF**	**AdaBoost**
1	138	88.889	88.889	82.222	78.889	68.889
2	67	84.444	85.556	82.222	75.556	71.111
3	36	86.666	80.000	80.000	74.444	71.111
4	13	81.111	76.667	78.889	76.667	72.222
5	10	78.889	80.000	75.556	72.222	73.333
6	8	75.556	81.111	80.000	76.667	77.778
7	4	74.444	78.889	73.333	71.111	71.111
8	2	73.333	74.445	71.111	66.667	63.333
9	2	73.333	74.445	71.111	66.667	63.333
10	2	73.333	74.445	71.111	66.667	63.333
11	1	72.222	68.889	62.222	44.444	50.000
12	1	72.222	68.889	62.222	44.444	50.000
13	1	72.222	68.889	62.222	44.444	50.000
14	1	72.222	68.889	62.222	44.444	50.000
**CPTAC Proteomic Dataset**						
**MGMT Known Status-Based FS**						
**LASSO = 2 and mRMR = 1**						
**k**	**# of Features**	**SVM**	**LR**	**KNN**	**RF**	**AdaBoost**
1	138	88.889	88.889	82.222	80.000	68.889
2	**114**	**90.000**	88.889	80.000	78.889	71.111
3	43	87.778	87.778	81.111	75.556	73.333
4	21	86.667	85.555	81.111	77.778	81.111
5	13	78.889	76.667	78.889	77.778	71.111
6	10	75.555	82.222	77.778	73.333	74.444
7	7	80.000	78.889	82.222	81.111	74.445
8	6	82.222	83.333	82.222	74.444	66.667
9	4	80.000	76.667	76.667	74.444	68.889
10	2	73.333	74.445	71.111	66.667	63.333
11	2	73.333	74.445	71.111	66.667	63.333
12	1	72.222	68.889	62.222	44.444	50.000
13	1	72.222	68.889	62.222	44.444	50.000
14	1	72.222	68.889	62.222	44.444	50.000

**Table 7 ijms-25-04082-t007:** The effects of feature selection methods in terms of mean squared error for 90 patients according to the MGMT protein expression level-based CPTAC dataset.

ML-MSE	Before FS	LASSO	mRMR
**SVR**	0.827	**0.772**	0.813
**RF**	0.867	0.908	0.886

**Table 8 ijms-25-04082-t008:** The performance results of the rank-based feature and selection method in terms of mean squared error for 90 patients according to the MGMT known status protein expression level-based CPTAC proteomic dataset. Bold values highlight the best outcome.

CPTAC Proteomic Dataset							
MGMT Known Status Protein Expression Level-Based FS							
LASSO = 1 and mRMR = 2				LASSO = 2 and mRMR = 1			
k	# of Features	SVR	RF	k	# of Features	SVR	RF
**1**	196	0.442	0.639	**1**	196	0.442	0.631
**2**	78	0.436	0.592	**2**	175	0.434	0.617
**3**	39	0.432	0.561	**3**	**56**	**0.406**	0.592
**4**	16	0.527	0.568	**4**	41	0.431	0.559
**5**	11	0.572	0.545	**5**	22	0.480	0.576
**6**	9	0.547	0.579	**6**	17	0.526	0.565
**7**	7	0.559	0.618	**7**	8	0.599	0.589
**8**	5	0.602	0.636	**8**	7	0.592	0.606
**9**	5	0.602	0.636	**9**	7	0.592	0.606
**10**	3	0.734	0.829	**10**	3	0.778	0.886
**11**	2	0.751	0.793	**11**	3	0.778	0.886
**12**	2	0.751	0.793	**12**	1	0.936	1.250

**Table 9 ijms-25-04082-t009:** Patient characteristics table for our local proteomic dataset.

Characteristics	N = 109	%
Sex		
Male	74	68%
Female	35	32%
Cortical/Periventricular		
Cortical	66	61%
Periventricular	43	39%
VPA		
Yes	31	28%
No	78	72%
MGMT Status		
methylated	27	24.77%
unmethylated	38	34.86%
unknown	44	40.37%
Age		
<35	3	3%
36–45	11	10%
46–55	32	29%
56–65	44	40%
66–75	18	17%
75+	1	1%
Type of Surgery		
Biopsy only	9	8%
STR	62	57%
GTR	37	34%
Unknown	1	1%
Steroid administration		
No record	14	12%
Before RT only	12	11%
During RT only	47	43%
Before and during RT	12	11%
No administration	25	23%
Days from surgery		
No record	3	3%
<20	13	12%
20–29	52	48%
30–39	30	27%
40–59	8	7%
60+ days	3	3%

**Table 10 ijms-25-04082-t010:** MGMT known status-based proteomic feature selection datasets employed for the classification tasks.

Dataset	Local	CPTAC
**The Number of Patients**	65	90
**The Number of MGMT Methylated Cases**	27	38
**The Number of MGMT Unmethylated Cases**	38	52
**The Number of Total Features**	7289	8838
**Cross-Validation Type**	5-Fold Stratified CV	
**Feature Selection Methods**	mRMR, LASSO for Classification	
**Feature Weighting Rule**	Rank-Based	
**Classifier Models**	SVM, LR, KNN, RF, AdaBoost	
**Performance Metric**	ACC	

**Table 11 ijms-25-04082-t011:** MGMT expression level-based proteomic feature selection datasets employed for the regression tasks.

Dataset	Local MGMT Known Status	Local MGMT All Status	CPTAC MGMT Known Status
**The Number of Patients**	65	109	90
**The Number of Total Features**	7288	7288	8837
**Cross-Validation Type**	5-Fold Stratified CV		
**Feature Selection Methods**	mRMR, LASSO with Regression		
**Feature Weighting Rule**	Rank-Based		
**Regression Models**	SVR, RF		
**Performance Metric**	MSE		

## Data Availability

De-identified data, including clinical data associated with the proteomic data set, will be shared once analyses for outcomes are complete.

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
