# Peer review of "MGMT ProFWise: Unlocking a New Application for Combined Feature Selection and the Rank-Based Weighting Method to Link MGMT Methylation Status to Serum Protein Expression in Patients with Glioblastoma"

_ijms, 2024, doi:10.3390/ijms25074082_

Round 1
Reviewer 1 Report
Comments and Suggestions for Authors
This study was addressed to non-invasively associate O6-methylguanine-DNA-methyltrasnferase (MGMT) promoter methylation status to serum protein expression in subjects with glioblastoma (GBM). Indeed, such status appears to represent an important biomarker for evaluating both response to temozolomide (TMZ), a classic therapeutical drug employed in GBM, and patient survival. Owing to the difficulty of obtaining tumor samples for MGMT testing, the present study adopted a combined feature selection (i.e. LASSO and mRMR) and a rank-based weighting method (i.e. MGMT ProFWise) for substantiating the above association between MGMT promoter methylation status and serum protein expression in GBM patients. Promising results were obtained when they were compared with two large-scale proteomic datasets (7k SOMAScan® panel and CPTAC). The computational evidences indicated that the chosen approach provided 14 shared serum biomarkers employable for a better clinic management of GBM-derived problems.
Introduction deals with biomolecular aspects pertaining to GBM and MGMT, including resistance to TMZ. Also technical and clinical aspects are detailed.
Materials and Methods describe Datasets, Methodology [Proposed Scheme, Feature Selection Methods (i.e. mRMR and LASSO), Feature Weighting Process, Classification and Regression (Support Vector Machine, Logistic Regression, K Nearest Neighbors, Random Forest and AdaBoost].
Results report Experimental Process, Performance Metrics and Computational Evidences (Local proteomic dataset-based results and CPTAC proteomic dataset-based results).
Discussion underlines the relevance of MGMT promoter methylation in both survival of GBM patients and their responsiveness to TMZ. In conclusion, Authors consider that future metabolomic and imaging features should be obtained to ameliorate the identification of non-invasive biomarkers in GBM. Discussion underlines the relevance of MGMT promoter methylation in both survival of GBM patients and their responsiveness to TMZ. In conclusion, Authors consider that future metabolomic and imaging features should be obtained to ameliorate the identification of non-invasive biomarkers in GBM.
As a marginal notation, one should consider that, although the methylating agent TMZ is widely employed in GBM pharmacopeia, its clinical efficacy is scant and, despite chemo- and radiotherapy, the median survival after diagnosis of GBM is in the order of 15 months. In this regard, new approaches in GBM chemotherapy are emerging, and particular attention is devoted to drugs acting against GBM stem cells (GSC, poorly responsive to chemo- and radiotherapy). Metformin and arsenic trioxide have been shown to exert lethal effects on GSC by blocking the pivotal LKB1/AMPK/mTOR/S6K1 pathway-dependent cell growth, by impairing the GSC-initiating spherogenesis, by inhibiting the proliferation of CD133+ cells, by inducing autophagy and apoptosis in glioma cells (through the PI3K/Akt and MAPK pathways), and by promoting differentiation of GSC into non-tumorigenic cells (by actions on the AMPK-FOXO3 axis and by blocking the IL-6-induced promotion of STAT3 phosphorilation). Molecular biology of GSC (within the stem cell theory of carcinogenesis) has been largely detailed, thus offering, as in the case of metformin and arsenic trioxide, a series of interesting targets for new drugs able to renew the obsolete chemotherapy of GBM with TMZ (see, for example: J. Biol. Regulators & Homeostatic Agents, 28, no. 1, 1-15, 2014).
The present study is somewhat original and interesting. It has been well planned and performed with scientific precision and appreciable complex and articulated methodologies. One may wish that the obtained evidences will be able to set the effects on GBM of drugs other than TMZ. Lexicon, sentence fluency, “English style”, references, tables and figures (including the supplementary ones) appear to be adequate.
Author Response
The authors appreciate the positive feedback from the reviewer. The authors also thank the reviewer for pointing out this constructive advice. The points made by the reviewer resonate with the authors and we collectively wanted to enhance the paper to reflect the points made regarding: 1) TMZ - we agree and have added content both on the limitations of TMZ as well as the need to link emerging data to GBM stem cells which are inherently radio and chemo resistant. 2) We agree that we need to discuss agents such as metformin and arsenic trioxide as well as HDAC inhibitors such as Valproic acid and other approaches that represent potential promising avenues in this space given the need to address and update therapy that is not as effective as we would have hoped. We have thus added references pertaining to the items above including also the reference mentioned by the reviewer.
This component was implemented in the introduction and now reads “This can be achieved by studying new approaches to systemic management including agents that act to potentially reverse or modify resistance by targeting GBM stem cells[15, 16], or modify response to RT[17] in an effort to link systemic management and RT to both MGMT dependent and MGMT independent mechanisms.”
The added references are:
- Carmignani, M., et al., Glioblastoma stem cells: a new target for metformin and arsenic trioxide. J Biol Regul Homeost Agents, 2014. 28(1): p. 1-15.
- Alves, A.L.V., et al., Role of glioblastoma stem cells in cancer therapeutic resistance: a perspective on antineoplastic agents from natural sources and chemical derivatives. Stem Cell Research & Therapy, 2021. 12(1): p. 206.
- Verdugo, E., I. Puerto, and M. Medina, An update on the molecular biology of glioblastoma, with clinical implications and progress in its treatment. Cancer Commun (Lond), 2022. 42(11): p. 1083-1111.
Reviewer 2 Report
Comments and Suggestions for Authors
Summary:
The manuscript titled "MGMT ProFWise: Integrating Feature Selection and Rank-Based Weighting Methods to Predict Serum Protein Biomarkers Associated with Known Molecular Markers in Glioblastoma Patients" presents a novel approach to linking MGMT methylation status with serum protein expression in GBM patients. The study aims to improve clinical outcomes by predicting protein biomarkers accurately, potentially leading to more tailored treatment approaches. The main contributions lie in the hybrid feature selection and weighting method employed, offering a focused selection of informative features for oncologic datasets. Congratulations to the authors for a well-written manuscript.
Comments:
- As part of the background, incorporate recent advancements in proteomics and their role in identifying potential serum protein biomarkers for GBM.
- Highlight the current gaps in knowledge regarding the association between MGMT methylation status, serum protein expression, and clinical outcomes in GBM.
- What is the rationale behind choosing LASSO and mRMR as the feature selection methods in the MGMT ProFWise approach? Also, provide the rationale behind selecting the specific machine learning models (SVM, LR, KNN, RF, AdaBoost).
- How were potential biases or confounding factors addressed? How was the stratified cross-validation approach utilized to account for class imbalances in the dataset?
- Elaborate on the clinical implications of the identified serum biomarkers for predicting MGMT status in GBM and address the feasibility of translating these findings into clinical practice as diagnostic or prognostic tools for GBM.
- Expand on the potential for further validation and clinical translation of the identified serum biomarkers, such as integrating them into trials.
- What is the rationale behind selecting the specific proteomic datasets used? How were the preprocessing steps tailored to the proteomic datasets and the specific requirements of the feature selection and modeling tasks? Provide further insights into how data preprocessing impacts the reliability of the study results.
Minor editing required, recommend authors to run a proof-reading software, such as Grammarly.
Author Response
Comment #1: As part of the background, incorporate recent advancements in proteomics and their role in identifying potential serum protein biomarkers for GBM.
Response #1: The authors thank the reviewer for highlighting this issue. We agree and have now enhanced the background to address advancements in proteomics and emerging patterns in enrichment analyses and serum biomarkers. We have added the following to the introduction section: “Recent advancements in proteomics whether they employed mass spectrometry and tumor tissue or large scale panels and serum biospecimens, have uncovered crucial relationships between GBM tumor heterogeneity[19], the proteome and the metabolome[20], the proteome and outcomes [21, 22] and observable footprints of alterations with concurrent management in proteomic data[23]. There is currently no serum biomarker in clinical use for GBM diagnosis or treatment monitoring however serum proteomic markers of promising importance are emerging in the literature and several have applicability to GBM[24]. “
The added references are:
- Lam, K.H.B., et al., Topographic mapping of the glioblastoma proteome reveals a triple-axis model of intra-tumoral heterogeneity. Nature Communications, 2022. 13(1): p. 116.
- Wang, L.-B., et al., Proteogenomic and metabolomic characterization of human glioblastoma. Cancer Cell, 2021. 39(4): p. 509-528.e20.
- Krauze, A.V., et al., Glioblastoma survival is associated with distinct proteomic alteration signatures post chemoirradiation in a large-scale proteomic panel. Frontiers in Oncology, 2023. 13.
- Yanovich-Arad, G., et al., Proteogenomics of glioblastoma associates molecular patterns with survival. Cell Rep, 2021. 34(9): p. 108787.
- Krauze, A.V., et al., Revisiting Concurrent Radiation Therapy, Temozolomide, and the Histone Deacetylase Inhibitor Valproic Acid for Patients with Glioblastoma-Proteomic Alteration and Comparison Analysis with the Standard-of-Care Chemoirradiation. Biomolecules, 2023. 13(10).
- Linhares, P., et al., Glioblastoma: Is There Any Blood Biomarker with True Clinical Relevance? Int J Mol Sci, 2020. 21(16).
Comment #2: Highlight the current gaps in knowledge regarding the association between MGMT methylation status, serum protein expression, and clinical outcomes in GBM.
Response #2: The authors thank the reviewer for pointing out this constructive advice. We agree. This is a very significant point.
We have now added to the introduction the following: “MGMT status is linked to response to TMZ which is standard of care administered concurrently with RT and adjuvantly, however this mainstay approach still results in very poor outcomes with a median survival of 14 months[9].”
And the associated Stupp reference:
- Stupp, R., et al., Radiotherapy plus concomitant and adjuvant temozolomide for glioblastoma. N Engl J Med, 2005. 352(10): p. 987-96.
To further address the gaps between MGMT methylation status, serum protein expression, and clinical outcomes in GBM we have also added the following to the introduction: “This can be achieved by studying new approaches to systemic management including agents that act to potentially reverse or modify resistance by targeting GBM stem cells[15, 16], or modify response to RT[17] in an effort to link systemic management and RT to both MGMT dependent and MGMT independent mechanisms.”
The added references are:
- Carmignani, M., et al., Glioblastoma stem cells: a new target for metformin and arsenic trioxide. J Biol Regul Homeost Agents, 2014. 28(1): p. 1-15.
- Alves, A.L.V., et al., Role of glioblastoma stem cells in cancer therapeutic resistance: a perspective on antineoplastic agents from natural sources and chemical derivatives. Stem Cell Research & Therapy, 2021. 12(1): p. 206.
- Verdugo, E., I. Puerto, and M. Medina, An update on the molecular biology of glioblastoma, with clinical implications and progress in its treatment. Cancer Commun (Lond), 2022. 42(11): p. 1083-1111.
Comment #3: What is the rationale behind choosing LASSO and mRMR as the feature selection methods in the MGMT ProFWise approach? Also, provide the rationale behind selecting the specific machine learning models (SVM, LR, KNN, RF, AdaBoost).
Response #3: The authors thank the reviewer for pointing out this constructive advice. We added the rationale and the related explanations regarding the selection of LASSO and mRMR feature selection methods and five supervised learning models (SVM, LR, KNN, RF, and AdaBoost) to the Section 4.2.1 Proposed Scheme as below:
“mRMR and LASSO methods are among the most commonly used and important algorithms for the feature selection process in the highly cited papers in the literature [1, 2]. In this study, we utilized the advantages (i.e., power) of both different types of feature selection methods (i.e., filter and embedded) by combining two different algorithms via rank-based weighting methodology. We also adopted five effective and popular machine learning classification models[3], including SVM, LR, KNN, RF, and AdaBoost, to assess the performance of our feature selection methodology.”
Additional References
- Fahimifar, S., et al., Identification of the most important external features of highly cited scholarly papers through 3 (ie, Ridge, Lasso, and Boruta) feature selection data mining methods: Identification of the most important external features of highly cited scholarly papers through 3 (ie, Ridge, Lasso, and Boruta) feature selection data mining methods. Quality & Quantity, 2023. 57(4): p. 3685-3712.
- Moslemi, A., A tutorial-based survey on feature selection: Recent advancements on feature selection. Engineering Applications of Artificial Intelligence, 2023. 126: p. 107136.
- Sarker, I.H., A. Kayes, and P. Watters, Effectiveness analysis of machine learning classification models for predicting personalized context-aware smartphone usage. Journal of Big Data, 2019. 6(1): p. 1-28.
Comment #4: How were potential biases or confounding factors addressed? How was the stratified cross-validation approach utilized to account for class imbalances in the dataset?
Response #4:
The authors thank the reviewer for pointing out this constructive advice. We added the related explanations to the Section 2.1 Experimental Process as follows:
“For the feature selection operation, potential biases and confounding factors may be selection bias (e.g., using the same data for the feature selection and classification), class imbalance, or feature interactions/correlations among them. To tackle these possible issues, we employed a stratified cross-validation technique to reduce bias by ensuring each fold has a similar ratio of classes as the original data, giving a more accurate view of the model's generalization ability across all labels. This approach provides a more reliable performance metric by ensuring the balanced spread of classes across folds and reflecting the model's performance on all classes. By ensuring each fold in cross-validation reflects the true proportion of classes in the data, stratified cross-validation guards against evaluation bias and prevents the model from being trained on an unrealistic distribution of classes. This approach also prevents skewed training of learning models. Due to unknown MGMT status cases, our stratified five-fold cross-validation technique corresponded to a five-fold cross-validation technique for only the local MGMT dataset for the regression task. We also utilized minimum redundancy and maximum relevance feature selection method to mitigate feature correlations-related bias factors.”
Comment #5: Elaborate on the clinical implications of the identified serum biomarkers for predicting MGMT status in GBM and address the feasibility of translating these findings into clinical practice as diagnostic or prognostic tools for GBM.
Response #5:
This is a very important point. We agree that addressing this is highly relevant and have now enhanced the discussion section by adding the following content: “Large scale proteomic panels are actively evolving to include larger repertoires of proteins thus as data sets grow and validation of findings improves, ranges can be generated for various GBM clinical feature sets and markers displayed in Figure 2B can be potentially employed to predict MGMT status. Currently MGMT status is determined based on tissue sample obtained at the time of surgical resection the analysis of which can be cost prohibitive and delay results which are often not available at the time of initial consult (2-3 weeks post resection). Serum markers can be measured with results returned within hours as is currently the case with markers such as CRP which is one of the markers identified in the current analysis. For treatment, monitoring panels can be finetuned further to home in on patterns that reflect likelihood of response or treatment failure, rendering the patient a candidate for clinical trial management. For example, serum markers that phenotypically reflect tumor behavior that mirrors the unmethylated patients in Figure 1, could be considered more optimal candidates for treatment with agents other than TMZ[15, 56].
Comment #6: Expand on the potential for further validation and clinical translation of the identified serum biomarkers, such as integrating them into trials.
We agree with the reviewer as this an active area research for us and subject to future directions. We have now expanded on this facet in the discussion by adding the following content: “Future directions of our research include validation and clinical translation of serum biomarkers into clinical trials. Validation is subject to comparative analyses with serum and plasma proteomic data sets several of which are currently evolving[34]. Clinical translation is contingent on implementation of serum markers in GBM trials. While over 30 trials are ongoing that aim to leverage biomarkers for the diagnosis or management of GBM[57], most involve tissue as the biospecimen of origin. A shift to liquid biopsies is needed, however a template for the real time measurement of novel serum markers needs to be established for analysis to occur in the clinic to direct management in real time and this is the subject of future directions.”
The following references were added:
- Candia, J., et al., Assessment of variability in the plasma 7k SomaScan proteomics assay. Scientific reports, 2022. 12(1): p. 17147.
- ClinicalTrials.gov. ClinicalTrials.gov. 2024 3/2024 [cited 2024 3/2024]; Available from: https://clinicaltrials.gov/search?cond=glioblastoma&term=biomarkers&page=5.
Comment #7: What is the rationale behind selecting the specific proteomic datasets used? How were the preprocessing steps tailored to the proteomic datasets and the specific requirements of the feature selection and modeling tasks? Provide further insights into how data preprocessing impacts the reliability of the study results.
Response #7:
The authors thank the reviewer for pointing out this constructive advice.
“Our first aim is to identify the relevant and significant biomarker sets regarding MGMT status or expression level from the large-scale proteomic dataset. To carry this out, we employed our local MGMT-based proteomic dataset. Given the paucity MGMT-based proteomic datasets, we also utilized CPTAC as a parallel proteomic dataset to validate our results and identify the shared biomarkers while methodologically approaching the similar tasks. The local proteomic dataset was the result of biospecimens collected on trial at our institution. The CPTAC proteomic dataset was selected for parallel analysis for the following reasons: 1) it is currently the single most accessible proteomic data set that is also linked to The Cancer Imaging archive; 2) it has MGMT status as an available feature; 3) it is the proteomic data set most popular in publications in this space. These facets were central to its selection as future directions include linkage to imaging features for our group, thus the selection of a set that was already linked to imaging data was central. We were equally keen to select a comparison set that could by linkage to serum data could most expeditiously advance evolving research given ease of accessibility and widespread use in the literature to eventually transfer findings.”
We have also added the following explanations to the Materials and Methods and 2.1 Experimental Process sections.
“Data preprocessing, including feature normalization, can significantly affect the effectiveness of feature selection techniques during the preprocessing stage for the prediction tasks. Features with high variance can be misleading for the feature selection algorithms. The feature normalization process helps mitigate this bias by focusing on the features that contribute meaningfully to the classification/regression task. In this study Somalogic data arrived normalized from the company by default. However, as the normalized data has a high variance, we additionally applied a logarithmic two-base transformation to reduce bias and provide more relevant feature selection. With respect to the CPTAC data set we adopted one of the most frequently applied data preprocessing approaches (i.e., z-score normalization). ”
Additional Explanations
A Grammarly check has been carried out, and corrections applied.